# Intercropping of Oats with Vetch Conducts to Improve Soil Bacteriome Diversity and Structure

**DOI:** 10.3390/microorganisms13050977

**Published:** 2025-04-24

**Authors:** Mariana Petkova, Stefan Shilev, Vanya Popova, Ivelina Neykova, Nikolay Minev

**Affiliations:** 1Department of Microbiology and Environmental Biotechnologies, Agricultural University—Plovdiv, 4000 Plovdiv, Bulgaria; mpetkova@au-plovdiv.bg (M.P.); ivelina.neykova@au-plovdiv.bg (I.N.); 2Department of Agrochemistry and Soil Science, Agricultural University—Plovdiv, 4000 Plovdiv, Bulgaria; nikiminev@abv.bg

**Keywords:** soil bacteriome, cereal–legumes intercropping, plant–microbe interactions, nitrogen cycling

## Abstract

Due to intensive crop cultivation, soil depletion has increased interest in intercropping, cover crops, and permanent land use. In our experiment, we investigated the effect of growing oats alone or together with vetch and green manure on the structure of the rhizosphere bacteriome. As a control, we used the bacteriome of bulk soil before sowing, during the growing season, and three months after green manure. The results revealed that its composition and structure change significantly due to the type of cultivation and the presence of plants. Plant roots had a significant impact by reducing pH and mobilizing nutrients. It was more evident in intercropping compared to the cultivation of oats alone. The rhizosphere bacteriome structure significantly differed from that of bulk soil. The different habitats in the rhizosphere and bulk soil led to a decrease in the representation of Proteobacteria and an increase in that of Actinobacteria. Intercropping contributed significantly to increased alpha diversity compared to the cultivation of oats alone and increased availability of nitrogen and potassium. The richness and inverse Simpson diversity index in the rhizosphere ranged from 7.361 to 8.015 across soils subjected to traditional oat cultivation or intercropping. However, the bacteriomes of the rhizosphere soil clustered together and were significantly different from those of the bulk soil. Our study provides a theoretical basis for understanding the influence of roots and cultivation type on bacteriome structure. It offers novel insights for soil management and plant health by reducing pathogens present in soil.

## 1. Introduction

Intercropping is the practice of growing two or more crops in the same field simultaneously [1,2]. The main objective of intercropping is to achieve a higher yield per unit area by utilizing resources or ecological processes that a single crop would not otherwise exploit. Combining annual forage crops to increase yields offers clear nutritional and financial benefits for livestock production. Intercropping winter cereals with annual legumes is widely practiced in many regions for forage production [3]. Oats (*Avena sativa* L.) are a well-known cereal crop in our country. The unique qualities of oat grain as fodder are due not only to its relatively high protein content (approximately 13%) but also to its significant fat content (4–6%) and the abundance of vitamins (B1, B2, B6, K1, A, and F). These levels exceed those found in other cereal crops [4]. Common vetch (*Vicia sativa* L.) is one of the most important forage plants. It is a nitrogen-fixing annual plant from the *Fabaceae* family [5,6]. In Bulgaria, common spring vetch is widely cultivated and is a key component of annual forage crops grown for hay [7]. Its mixture with oats is the primary source of high-quality hay in grain-producing regions. Vetch is drought-tolerant and the most cold-hardy among winter annual legumes [6]. It is an excellent nitrogen fixer due to its symbiosis with nitrogen-fixing bacteria (*Rhizobium* sp.), and it demonstrates strong competitiveness against weeds. Its robust root system makes it an excellent partner for intercropping with cover crops to improve soil structure and rapidly produce biomass [8]. Vetch’s deep roots also help prevent soil compaction, common in monoculture systems. Vetch competes effectively with weeds, further improving the efficiency of the cropping system [9]. Beyond its nutritional value, oats play a crucial role in the intercropping system as a structural support for vetch. Vetch, a climbing legume, uses oats as a support system, reducing matting and enhancing the incorporation of both crops into the soil. The physical presence of oats helps improve soil coverage, reducing erosion and increasing moisture retention, which can enhance the growing environment for both plants. Furthermore, oats’ shading effect helps suppress weed growth, a common issue in monoculture systems [10].

When grown alongside oats, vetch uses the oat plants as a support to climb during spring, which reduces matting and improves its incorporation into the soil [11]. The advantages of intercropping these crops include higher forage quality due to increased crude protein concentrations in legumes, enhanced biomass production [12], reduced nitrogen fertilizer usage, and the production of higher-quality animal feed. Such practices can improve the quantity and quality of forage compared to monoculture cereal crops [13]. Compound feed is a common agricultural practice used to meet livestock’s energy and protein needs. Marković and coworkers demonstrated the positive effects of oats–vetch intercropping by comparing the independent cultivation of oats and vetch to their mixed cultivation at different sowing ratios [14]. Similarly, Pużyńska investigated common vetch and oats mixtures at varying sowing rates [10]. The results showed that the intercropping of vetch and oats at different ratios significantly influenced fresh and dry matter yield.

Studies have shown that intercropping oats and vetch produces higher biomass than monoculture cereal crops, as the two species efficiently utilize different ecological niches [15]. Vetch fixes nitrogen, while oats provide physical support, contributing to a higher total biomass. Combining oats’ high-quality grain and vetch’s nitrogen fixation results in a higher protein concentration in the forage. It increases the nutritional value of the resulting hay, making it more beneficial for livestock [16]. Vetch’s nitrogen-fixing ability reduces the need for synthetic fertilizers, lowering production costs and minimizing the environmental impact of fertilizer use. It contributes to a more sustainable farming system [17]. Thus, the intercropping system enhances soil structure by combining oats’ root systems and vetch’s nitrogen fixation, creating a healthier, more resilient soil ecosystem, supporting higher yields over time [18].

Different ecological niches host diverse microorganisms. Plants alter the biotic properties of soil, and these changes, in turn, affect plant growth, survival, and reproduction [19]. However, the reverse interactions, particularly the mechanisms involving invasive plant species, remain poorly understood. Since only a limited number of microorganisms in natural ecosystems can be cultivated in a laboratory, traditional microbiological methods are insufficient for revealing the true species diversity and the dynamics of mixed cultures associated with biochemical changes in soil. A study investigated soils from an oat–vetch intercropping system to evaluate their potential to achieve higher yields by utilizing resources and ecological processes unavailable to single crops. The results were comparable with previous studies on camelina–vetch intercropping [20]. Oat–vetch intercropping represents a sustainable farming method that enhances productivity and conserves soil health. However, the success of such a system depends on specific agricultural and regional conditions, underscoring the importance of soil estate assessments before planting. When combined in an intercropping system, oats and vetch exhibit a synergistic effect: oats provide structural support and soil shading, while vetch enriches the soil with nitrogen. In another study related to the influence of different systems of fertilization and liming on the changes in the taxonomic structure of prokaryotic community, the authors reported the following representation of phyla depending on the treatments: *Proteobacteria*, (45.3–56.2%), *Actinobacteria*, (13.6–20.4%), *Bacteroidetes* (7.2–19.3%), *Acidobacteria* (7.1–11.5%), and *Verrucomicrobia* (4.3–10.3%) [21]. They found a higher amount of the most represented phyla—*Proteobacteria*—in the treatment with black fallow. Unfortunately, they did not investigate the influence of intercropping on the structure of bacteriomes. The soil bacteriome and the microbiome, as a whole, are very important for soil health because of the multiple functions and the number of agroecosystem services they realize. In the present literature revision, we found considerable gaps related to the lack of sufficient studies concerning the influence of intercropping on soil rhizosphere bacteriome. Therefore, the main objective of our study and novelty was to investigate how the intercropping of oats and vetch influences the oat’s rhizosphere bacteriome structure. The secondary objective was related to the study of variation in soil bacteriome in unplanted soils as controls.

## 2. Materials and Methods

### 2.1. Field Description, Experimental Design and Soil Sampling

The experiment was carried out at the Training and Experimental Field of the Agricultural University of Plovdiv on alluvial–meadow soil (42°08′15.4″ N 24°48′16.4″ E). The soil in the area is characterized by huge diversity in terms of its type. The alluvial–meadow soil is formed on alluvial deposits and has a well-formed humus-accumulative horizon, which gradually passes into the carbon horizon. A gleization process is observed deeply down (below 100 cm) in the soil-forming material—the A-C-G profile. At the same time, according to the International Classification of FAO, it refers to Mollic fluvisol. We applied a randomized block design with three treatments with three replicates each: control filed (fallow land, uncultivated), oats (*Avena sativa* L., variety Max), and oats–vetch intercropping (*Vicia sativa* L. variety Obrazets 666). Each replicate was located on a plot of 35 m^2^, leading to a total of 105 m^2^ per treatment. The plants were sown in mid-March of 2022: oats at 170 kg/ha, while in the intercropping, the seeds were planted at 220 kg/ha (3:1, vetch/oats). Fertilization was carried out in the autumn of 2021 with 200 kg/ha NPK (15:15:15) before basic tillage. At the end of June 2022, the oats were harvested, and the oats–vetch mixture was incorporated into the soil at 20 cm as green manure.

Rhizosphere soil samples were taken from each of the three replicates from 0–20 cm depth on six occasions: before sowing at the beginning of March 2022 (S1—control field), from plant rhizosphere at ripening phase in mid-June (S2—control filed, S3—oats and S4—oats intercropping with vetch) and post-harvest in mid-October (S5—soil after intercropping and green manuring; S6—control field) (Table 1). The sampling plants in the corresponding replicate were randomly selected at each sampling time. The resulting eighteen samples were designated for soil rhizosphere high-throughput amplicon sequencing and all samples were processed immediately. 

### 2.2. Soil Nutrient Analyses

The samples were dispersed in deionized water (1:10, *w*/*v*) and shaken (30 min), followed by sedimentation (10 min) and measurement of pH and EC at laboratory pH-EC-meter [22]. Total nitrogen was assessed using the Kjeldahl method. The ammonia (NH_4_-N) and nitrate (NO_3_-N) nitrogen were extracted using 2M KCl for 1 h. After filtration, the samples were determined spectrophotometrically [23]. Extractable phosphorus was evaluated using the method of ammonium molybdate [24], while potassium was assessed after extraction with 1N HCl and measurement at PFP-7 flame photometer (Cole-Parmer, Vernon Hills, IL, USA) [25]. Organic C was measured and calculated using the potassium dichromate method [26,27].

### 2.3. High-Throughput Amplicon Sequencing

#### 2.3.1. DNA Extraction

For microbiome analysis, soil samples were collected from the experimental fields at the Agricultural University of Plovdiv in 2022 (Appendix A). Each DNA sample was extracted and purified from 10 g of rhizosphere soils using a standard method as described by Chiodi et al. [28]. The quality of genomic DNA (gDNA) was assessed on a 1% agarose gel, running 5 µL samples at 70 V for 60 min to ensure a single intact band. The targeted regions were PCR-amplified using specific primers with barcodes for saline soil. The V4 region of the 16S rRNA gene was amplified with primers V3 (5′-CCTACGGGNGGCWGCAG-3′) and V4 (5′-GACTACHVGGGTATCTAATCC-3′) [29]. The precise amount of PCR products from each sample was combined, end-repaired, A-tailed, and then ligated with Illumina adapters. The libraries were sequenced on an Illumina paired-end platform, producing 570–590 bp paired-end raw reads. These reads were merged and pre-processed to obtain Clean Tags [30]. Chimeric sequences were detected and removed to produce Effective Tags for further analysis, following the method of Krstić Tomić et al. [31].

High-throughput amplicon sequencing was performed at Novogene (Cambridge, UK), with library preparation using the Nextera DNA Flex kit (Illumina, Cambridge, UK) according to standard protocols. The amplicon was sequenced on an Illumina paired-end platform, producing 570 bp paired-end raw reads, which were then merged and processed to obtain clean tags. Data were analyzed using QIIME software, version 1.9.1 [32]. Sequences with ≥97% similarity were assigned to the same OTUs. For each representative sequence, the Silva Database (http://www.arb-silva.de/, accessed on 8 January 2025) → (Quast C et al.) was used based on the Mothur algorithm to annotate taxonomic information [29]. According to Zhang et al. [33], to study the phylogenetic relationship of different OTUs, and the difference in the dominant species in different samples (groups), multiple sequence alignments were conducted using the MUSCLE software (Version 3.8.31, http://www.drive5.com/muscle/, accessed on 9 January 2025) [34]. Ternary plot and Venn and Flower diagram were performed using R software.

#### 2.3.2. Alpha Diversity

Alpha diversity metrics summarize an ecological community structure by assessing the number of taxonomic groups and their abundances, as described in [35]. To analyze alpha diversity, six indices, including Observed-species and Shannon, were calculated using QIIME (Version 1.9.1, http://qiime.org/) and visualized with R software (Version 2.15.3) [32,34]. Heatmaps based on weighted Unifrac and unweighted Unifrac distances were also generated with R software (Version 2.15.3). R software was utilized to identify differences in dominant taxa across three sample groups at each taxonomic level, and the top 10 and 30 taxa with the highest average abundance were selected for a ternary plot.

#### 2.3.3. Beta Diversity

During the 16S sequencing analysis, the beta diversity distance matrix, comprising the OTU abundance differences between two samples, was used as the input file. A probability level of less than 95% (0.05) was calculated [36]. Beta diversity, a comparative analysis of microbial community composition, was performed using taxonomy annotation results and abundance data from all samples [37]. This analysis evaluates differences between samples based on weighted and unweighted Unifrac metrics, calculated using QIIME software (Version 1.9.1) [37,38]. Differences between sample groups were identified through a Beta diversity index inter-group difference analysis via principal component analysis (PCA), utilizing the FactoMineR and ggplot2 packages in R software (Version 2.15.3) [39]. Principal coordinate analysis (PCoA) was used to derive principal coordinates and visualize complex multidimensional data, transforming the distance matrix of weighted or unweighted Unifrac among samples into a new set of orthogonal axes. The first principal coordinate represents the most significant variation factor, followed by the second greatest, and so on. PCoA analysis was visualized using the WGCNA, stat, and ggplot2 packages in R software (Version 2.15.3) [40]. Additionally, the unweighted pair-group method with arithmetic means (UPGMA) clustering was employed to interpret the distance matrix using average linkage, conducted with QIIME software (Version 1.9.1) [41]. In this clustering method, samples with the closest distances are grouped to form new nodes, with the average distance calculated between the most abundant and other samples.

#### 2.3.4. Data Availability

The 16S rRNA sequencing data from intercropped soils with oat and vetch were submitted to the NCBI gene bank under the BioProject accession number PRJNA1190667 with the submission ID: SUB14872245 released on 26 November 2024. The BioSample accessions of the analyzed six samples are SAMN45052725, SAMN45052726, SAMN45052727, SAMN45052728, SAMN45052729, and SAMN45052730. The data records are accessible using the following link: https://www.ncbi.nlm.nih.gov/sra/PRJNA1190667, accessed on 21 April 2025.

### 2.4. Soil Data Analysis

One-way analysis of variance (ANOVA) was carried out for all variables at a probability level of *p* ≤ 0.05. The significance of differences between mean values of the studied parameters was assessed through the least significant difference (LSD) test.

## 3. Results

### 3.1. Soil Physicochemical Parameters

Soil analyses before sowing showed low TOC—11.21 g/kg (Table 2). Soil pH was slightly alkaline, while the electrical conductivity was relatively low. We found a generally low concentration of available nutrient content in the soil; thus, it was poorly supplied with accessible nitrogen and well supplied with mobile P and K. The soil pH values in treatments with plants (S3–S5) were lower than those of the controls, probably due to the enhanced microbial activity and root exudates (Table 2). Moreover, EC was affected by the type of treatment. In this regard, the intercropping case value was much higher than that of the oats cultivated alone. This is due to the combined effect of vetch roots and the indirect effect of improved microbiome diversity and activity. Total N was increased in the intercropping treatment compared to the monoculture oats, which was reflected in both components—ammonium and nitrate N. In continuation, the soil concentration of P was 37.9% lower in the mixed cropping compared to the oats alone, which could be associated with P’s need for vetch and enhanced absorption in mixed cultivation and the limited source of P. Potassium in soil was found to be the highest before sowing. The concentration of this element was 28% higher in intercropping than in the treatments of oats in monoculture.

### 3.2. High-Throughput Amplicon Sequencing of Soil Bacteriome

#### 3.2.1. Sequencing Results

Table 3 summarizes the sequencing results from soil samples collected during different stages of crop growth and management practices and highlights the influence of various agricultural practices on soil microbial composition. In constructing OTUs, a variety of data were collected from the different samples, such as RawPE, Base, and tags annotation data (Table 3). The Illumina Miseq sequencing of the three samples produced 542,870 raw tags. After quality control, 529,467 clean tags remained. After removing chimeras, effective tags known as nochime were obtained, ranging from 104,163 for S1 to 119,471 for S3 for OTU generation. S4, S5, and S6 had the highest observed microbial count and exhibited greater microbial diversity than other sites. The Q20 values for the three samples ranged from 92.86% to 94.31%, demonstrating high-quality Illumina sequencing. The remaining sequences (unique tags: 5009) did not match any known bacterial sequences in the public database. These results suggest that the sequencing depth adequately captured many bacterial species. After removing the chimeras, practical markers were obtained to generate OTUs with an efficiency of 80.22–83.53% (Table 3).

#### 3.2.2. Bacterial Diversity and Abundance Curves

The abundance of the major bacterial groups in each taxonomic category is given in Figure 1A. *Proteobacte9ria* (25–37.5%) was the most dominant group, followed by unidentified bacteria (18–20%), *Actinobacteria* (6–13%), *Actinobacteriota* (7–8.5%), Acidobacteriona (2.5–7%) and *Firmicutes* (2–5%), (Figure 2). At the phylum level, 10 out of 135 phyla were common to all samples. Additionally, the microbial composition of these ten dominant phyla differed significantly (*p* < 0.05) between the six soils taken at different stages during the vegetation (Figure 1A). The highest abundance of *Proteobacteria* was observed in the control fields: before sowing (S1), ripening phase (S2), and after green manure (S6). Their contribution to the microbiome structure decreased in the rest of the treatments, which are treatments with plants (S3, S4, S5). This could be a result of a reduced portion of these communities among the bacteriome. Thus, Proteobacteria could be considered autochthonous in our soils, successfully surviving without plants. The phylum *Proteobacteria* includes several classes that involve plant growth and effects as biological control agents for various diseases [42]. In addition, the contribution to *Actinocateriota* in the bacteriome is higher in soil with plants (S3–S5), suggesting the importance of rhizosphere to microbiota development and structure.

At the genus level, we found abundance of *Sphingomonas* in the studied soils decreasing in the sites with plants (Figure 1B). In the same way, the presence of *Ralstonia* sp. significantly decreased in the soils of intercropping and especially after green manure, where an increase of the abundance of *Bacillus* was also found. In contrast, *Skermanella* sp., a genus of *Proteobacteria* phylum was even more increased in the intercropping soils three months after green manuring (S4 and S5).

*Cyanobacteria* 3%, *Ralstonia picketti*, and *Bacteriacoccus minor* were found in the rhizosphere soil of pre-sowing control, S1 (Figure 1C), gradually reduced the ripening time of oats and vetch, and were not observed in October in the field after intercropping. Bacterial species similar to the control S1 soil occur in the rhizosphere soil of oats–vetch. *Peanibacillus ssp*, *Bradyrhizobium,* and *Microvirga ossetica* were found in soil with oat and vetch co-culture (S4) and green manure soil (S5). Firmicutes are the metabolically most versatile group that realizes multiple enzymatic activities [43]. The present investigation found that in the rhizosphere soil *Bacillus* sp. and *Paenibacillus* sp. belong to Firmicutes.

The refraction curves of bacterial diversity showed different abundances of bacterial species (Appendix A). S4 had the highest abundance of prokaryotic species, while the pre-sowing control soil S1 had the lowest. S2, S3, and S5 have rarefaction curves; therefore, the microbial community composition in the six samples is similar. Rarefaction curves were collected, showing the number of operational taxonomic units (OTUs) clustered at a 97% sequence similarity cut-off using the Mothur program [44].

A total of 35 types of bacteria defined the microbiome of rhizosphere soils sampled during sole and co-cultivation of oats and vetch (Figure 2). *Bdellovbribnota* and *Cyanobacteria* predominated in the control soil. In self-grown oats, *Kryptonia* and *Spirochaetota* are found in greater quantity. The greatest richness of melts was found in the joint cultivation of the two cultures *Acidobacteriodota*, *Synergistota*, *Latescibacterota*, *Methylomirabilota*, *NB1-J*, and *Latescibacteriota* (Figure 2). In S4 soil, some bacteria have shown a role in intercropping *Methylomirabilota, Deltaproteobacterial* order *NB1-J*, *Latescibacteriota*, *but also Chloflexi*, *Entotheeonellaeota*, *Desulfobacteriota*, and *Planctomycetota*.

#### 3.2.3. Ternary Plot and Venn Diagram

The position of the circle represented in the ternary plot in Figure 3A shows the contribution of the bacterium to differentiating the composition of the caste microbiota. The most abundant species belong to *Ralsonia pickettiii, mostly found in the soil without plants*, while *Glutamicibacter arilaitensis* and *Bradyrhizobium elkanii* were common in sole cultivation of oats (S3) and in intercropped soil (S4). *Stenotrophomonas* sp. and *Sporosarcina psychrophila* appeared only in intercropping.

The diagram in Figure 3B focuses on the shared core and the unique features, but it does not explicitly display pairwise overlaps between groups (e.g., features shared only between S1 and S2). The core features of 214 common species form a significant overlap among all S1 to S6 soils. The intercropping (S4) supported microbial (prokaryotic) diversity compared to the oats’ bacteriome (S3) with 30 unique species. Moreover, green manure (S5) soil had the highest unique features (183), with 17 more than that of intercropping, while S1 has the lowest (81). This could suggest more outstanding distinctiveness or specificity in the S5 soil microbiome compared to the others.

#### 3.2.4. Alpha Diversity

In microbial ecology, analyzing the alpha diversity of amplicon sequencing data is a common first approach to assess differences between environments (Willis, 2019 [35]). Diversity indices (Simpson and Shannon) and richness indices (Ace and Chao1) were calculated to estimate the diversity of the bacterial communities in studied soils (Table 4).

Table 4 shows the richness and diversity of bacterial communities in S1 to S6. There are some differences in diversity indices between the samples. The lowest number of observed prokaryotic species was found in the control soil in March in the beginning of the season and before sowing (3307). During the vegetation period, this number increased in June (5.3%) and October (10.8%). There was no significant difference among the numbers of the control soil and oat rhizosphere in June and green manure soil in October, while intercropping contributed to a considerable increase in species abundance, with almost 14% compared to the oats and 15% compared to the control. After green manuring, species decreased by 13% in autumn. The highest bacterial diversity was observed in the control soil in autumn, 7.1%% higher than the data at the beginning of the spring (S1). The functional prokaryotic diversity had similar trends to those of the observed species. However, no difference was found in the Shannon diversity index data between rhizosphere samples of oats alone or with vetch, probably due to the increased richness of oats communities. The contribution of vetch as a leguminous plant to the diversity of soil bacteriome is also evident in Chao1 and ACE indexes. All these data suggest that the proportion of some dominant species is higher when intercropping is applied (Table 3).

#### 3.2.5. UniFrac-Weighted and Unweighted Analyses

Beta diversity analysis measures the differences in microbial communities between samples. It helps to assess the variation in microbial composition across different soil treatments or conditions, indicating how similar or different the microbial communities are from one another. Unlike alpha diversity, which measures within-sample diversity, beta diversity focuses on the differentiation or similarity between samples. Beta-diversity and PCA were also performed to determine differences and similarities in the distribution of prokaryotes among all soil samples by cluster analysis (Figure 4).

The result of UniFrac-weighted analysis showed the lowest beta diversity difference between the bacteriome of control fields before sowing and during ripening (S1 and S2, 0.169), but also between S2 and S6, suggesting high similarity of control fields and low importance of seasonal variation. A similar difference was observed between the oats’ bacteriome (S3) and the one in the oats–vetch intercropping rhizosphere (S4) during the ripening phase. The highest difference in the bacteriome’ structure was found between both of the control fields (S1 and S2) with intercropping and green manure soils (S4 and S5). Root exudates and crop type appeared to have a significant role in shaping the soil rhizosphere bacteriome, altering its structure. According to the UniFrac-unweighted analysis, the bacterial diversity of S1 was 0.397 with S2, and 0.413 with S6. The diversity of S4 was 0.370 with S3, and 0.402 with S5. When comparing beta diversity during vegetation (June) between unplanted (S2) and rhizosphere soil, we found 0.363 with the oats’ bacteriome (S3) and 0.344 with the intercropping one (S4) (Figure 4).

#### 3.2.6. Unweighted Group Pair Method with Arithmetic Mean (UPGMA)

Control fields without crops, where Proteobacteria and Actinobacteria prevail, have a minor role in the degradation processes (Figure 5) and cluster together. A separate cluster obtained by the unweighted pair method included bacteria in soils with sole or co-cultivation of oat and vetch S3, S4, and S5 clustered together. Similar to the results of Figure 2 and Figure 3, the most abandoned species belongs to Proteobacteria in the second cluster.

The Unifrac distance and the unweighted Unifrac distance were chosen to measure the difference coefficient between pairwise samples, a widely used phylogenetic measurement method in microbial community sequencing projects. Cluster analysis was used to construct a cluster tree in Figure 6 to examine the similarity between different samples.

#### 3.2.7. Principal Component Analysis (PCA)

PCA analysis can extract two coordinate axes that reflect the difference between samples to the greatest extent, summarizing the data on the two-dimensional coordinate map. The more similar the composition of the samples is, the smaller their distance in the PCA map. The results of PCA analysis based on feature consistency level are shown in Figure 6. The results showed a significant percentage of the total variation in the bacterial microflora, with 22.32% on the abscissa and 27.61% on the ordinate between all six soil samples S1–S6. In our case, S1 and S2 form one cluster, which is negatively related to both components (the red ellipse). Samples S3 and S4 are generally separated and negatively correlated with PC2, but they are closely situated and positively related to PC1 (the green ellipse). This cluster demonstrated similarity between the bacteriomes of oats intercropping. The structure of sample S5 seems different from the other clusters and suggests another type of bacteriome related closely to the PC1. It is more likely composed of communities with degrading abilities developed after green manuring the mixed crops.

#### 3.2.8. Principal Coordinate Analysis (PCoA)

The PCoA plots provided display the relationships among six treatments based on two distance metrics, weighted UniFrac and unweighted UniFrac. The *x*-axis represents PC1, capturing the most significant percentage of variance (54% for weighted UniFrac, 30.13% for unweighted UniFrac, Figure 7a). The *y*-axis represents PC2, accounting for the following most considerable variance (27.71% for weighted UniFrac, 22.09% for unweighted UniFrac). This plot considers abundance information in the samples where S1 (red) and S2 (blue) appear the farthest apart, indicating significant differences in community structure and abundance between these groups. On the other hand, S4, S5, and S6 (rhizosphere samples) cluster relatively close to the soil bacteriomes, suggesting similar microbial compositions with shared abundance distributions. Considering the unweighted UniFrac (Figure 7b), microbiome before sowing (S1) is separated from the different groups along PC1 and PC2, showing a distinct microbial composition. S2 remains an outlier on the negative PC2 axis, confirming distinctiveness regarding unique features, but is close to S3 at PC1. The bacteriomes of intercropping and those after green manure are very similar according PC1 and PC2, suggesting a more similar feature presence.

## 4. Discussion

Soil microbiome structure and diversity, as well as the bacteriome one, influence soil and plant health differently [45]. In our investigation, we studied the influence of intercropping with vetch on the soil bacteriome structure of oats and the changes compared to the control (unplanted) soils. Common vetch is important in organic farming due to its high protein content and low soil and climatic requirements. In intercropping soil, we found higher concentrations of most of the studied elements in available form compared to the treatment with oats’ cultivation. We think this is due to the combined cultivation and the contribution of vetch as a leguminous plant. A lower concentration of P was found, which may be connected to the increased EC and increased demand for P for both crops. As P occurs in low-solubility forms in soils, intercropping may lead to enhanced root exudates followed by increased P bioavailability in soil [46]. Other researchers found that intercropping oats with vetch did not significantly increase the soil C content in the short term, which confirms our findings [46,47]. The C:N ratio in the soil is significant for C storage. The return of organic matter with a low C:N ratio leads to increased C accumulation in soil due to microorganism requirements [48]. Our study found increased soil TOC due to green manuring. Contrarily, additional N fertilization will stimulate C loss from the soil due to accelerated organic decomposition [46]. Legume cultivation is known to increase soil N content [49]. In our study, we saw some increase in cases of intercropping that could be due to the legume plant association with soil N2-fixers. It is controversial, as in other studies, that it has not been reported [46]. Moreover, dissolved N and organic C concentrations are of considerable importance for the shift in microbial activity [50]. Studies utilizing high-throughput 16S rRNA sequencing have demonstrated that microbial communities are susceptible to varying land use practices [51]. Our study explored how microbial communities differ depending on the cultivation and its type. We found that green manure and intercropping significantly affect soil microorganisms’ structure, diversity, and richness. Alpha-diversity and beta-diversity analyses were conducted on all cases to assess microbial diversity, abundance, and distribution. Variation analysis revealed intercropping as a primary driving force in alpha and beta diversity, reaching the highest observed species count (4003 species) in S4 samples. These findings align with the results of other researchers, demonstrating the strong contribution of vetch to the soil microbiome structure and indicating a balanced and enriched microbial environment [52]. The suggestion is that intercropping promotes microbial diversity, possibly due to the increased availability of nutrients and root exudates from diverse plant species. Our understanding is that these differences resulted from the changes in physicochemical properties, thus contributing to the improved bacteriome biodiversity and modified community structure. Despite nutrient significance, pH and liming are important in determining the microbiome’s performance in intercropping oats and vetch. The poorer the soil, the greater the importance of fertilizers and liming for the structure of bacteriome [21,53]. Oats–vetch intercropping increases the Shannon diversity index and even beta diversity, enriching the rhizosphere with additional substances [54]. The results indicated that sites S3, S4, and S6 exhibited high microbial community diversity, with Shannon diversity indices of 9.590, 9.758, and 9.591, respectively (Table 4). By cultivating multiple crops together, the intercropping approach leverages the benefits of diverse plant species working synergistically [55,56]. The microbial communities across the six sites (S1–S6) exhibited distinct variations. Sites with similar land-use practices or salinity levels (e.g., S1–S5) clustered closely, indicating shared microbial composition. Site S6, which was highly distinct, suggests a unique microbial community structure potentially driven by specific environmental factors or intercropping effects. The distinct separation of S6 in the PCA plot highlights significant beta diversity, likely due to unique soil characteristics or intercropping influences. The unweighted pair group method (UPGMA) clustering revealed that the bacterial communities in soils with oat and vetch co-cultivation (S3, S4, S5) clustered together [52,53]. This suggests that intercropping systems, which introduce plant diversity and a more complex set of root exudates, create a distinct ecological environment that promotes a specific microbial community structure. In contrast, control soils without crops (S1) and soils with only oats (S2) showed different clustering patterns, highlighting the impact of plant type and cropping systems on microbial community composition. This cluster analysis supports the idea that intercropping systems, such as oat–vetch, foster greater microbial diversity and alter community structure in ways that contribute to enhanced soil fertility, nutrient cycling, and plant health. Intercropping enhances microbial diversity by promoting a heterogeneous soil environment, where diverse root exudates and organic inputs create niches for various microbial species. This microbial richness and evenness increase supports key ecological functions, including nutrient cycling and pathogen suppression. The data suggest that intercropping may increase the dominance of certain microbial groups adapted to these enriched environments. These dominant species contribute significantly to nutrient turnover, soil health, and plant growth. Intercropping systems enhance soil nutrient availability, particularly nitrogen (N) and phosphorus (P), and foster diverse microbial communities, as evidenced by numerous studies [57]. Findings from experimental plots suggested that sites with similar salinity levels had comparable microbial ecosystem distributions, aligning with the work of Yang et al. (2017) [58], who reported that salinity significantly shapes microbial community structure. Site S4 showed the richest microbial composition, as indicated by the Chao1 index, reflecting a greater variety of microbial species than other sites (Table 4). Principal component analysis (PCA), a component of beta-diversity analysis, revealed that microbial compositions from S1 to S5 were similar, whereas S6 was distinct from the others (Figure 5 and Figure 6). Beta diversity reflects ecological adaptability and the potential for microbial communities to respond to environmental changes. Higher beta diversity values of bacteriome among cultivated and intercropped rhizosphere indicate different species composition with consequences on functional diversity [59]. These higher differences are the distance of community composition among the control samples and the cultivated samples in the microbial communities [60]. The application of the different types of crops, such as leguminous and especially vetch, results in a higher beta distance with cereal ones [52]. The authors observed a clear separation of vetch position in the Bray–Curtis distance, suggesting it is due to the alteration in nitrogen turnover. In our study, the cluster grouping was based on the type of samples—rhizosphere samples or bulk soils (Figure 6). Bacteriomes of single oats cultivation and intercropped with vetch showed the same grouping as that of closely situated green manure bacteriomes. On the other hand, the bacteriome structure of un-vegetated samples was grouped apart, summarizing that the relative weight of Proteobacteria was lower in rhizosphere samples than in the non-rhizosphere ones. In this regard, the presence of Actinobacteriota, Chloroflexi and Crenarchaeota was increased. Different results were observed in the case of Chinese cabbage, where Proteobacteria were more abundant and had more representation, while Chloroflexi was in the lower portion of the soil bacteriome structure of rhizosphere than in the bulk soil [61]. Dominant phyla such as Proteobacteria, Actinobacteria, and Firmicutes were prevalent at the genus and species levels, highlighting their evolutionary adaptation mechanisms (Figure 6). Proteobacteria and Actinobacteria contributed significantly to nitrogen metabolism and organic matter mineralization (Figure 5 and Figure 6). Representatives of these phylums produce extracellular enzymes that convert organic matter into soluble forms of essential nutrients like phosphorus, nitrogen, and potassium [62]. Additionally, several bacteria, such as *Nitrospira*, *Deferribacteres*, and *Verrucomicrobiota*, thrived in saline conditions, indicating their adaptive capacities [63]. Thus, in our experiment, the concentration of organic C in the rhizosphere soil of intercropping during ripening was decreased compared to that in oats’ rhizosphere and the green manure stage. We suggest that this is because of available nitrogen in the soil that supports organic matter decomposition by microbiota. The genus *Bacillus*, known for its spore-forming capabilities and resilience in saline soils, was prevalent across the sites. The *Bacillus* species enhance plant growth, produce industrially significant enzymes, and participate in bioremediation [64,65]. Similarly, *Sphingomonas*, a genus within Proteobacteria, was present across all sites but most abundant in S6. This genus obviously decreases in abundance in rhizosphere soil, suggesting the existence of strong competition among species for the root exudates. We also found *Skermanella* sp. to be more abundant in the intercropping site that in the sole cultivation of oats or in controls. This suggests a correlation with the presence of vetch in the field. The genus *Skermanella* belongs to the family *Azospirillaceae*, well-known free-living nitrogen fixers [66], and is phylogenetically closely situated to the diazotroph *Azospirillum* [67]. This genus is dominant in different plants’ rhizosphere, such as cucumber, tobacco, and grapevine possessing *nifH*-gene-encoding reductase of nitrogenase [68]. These bacteria contribute to nutrient cycling, organic pollutant degradation, and plant health through phytohormone production and pathogen suppression. Moreover, Actinobacteria are indeed well known for their ability to break down complex organic materials, particularly through the secretion of extracellular enzymes that decompose plant polymers such as cellulose, hemicellulose, and lignin. These bacteria play a critical role in organic matter decomposition, nutrient cycling, and soil fertility [69,70].

The intercropping of oats with vetch supports sustainable farming practices by boosting soil nutrient availability, enhancing microbial activity, and improving crop yield and quality. It also contributes to ecosystem resilience under challenging agroecological conditions. The weighted UniFrac plot emphasized abundance differences, while the unweighted UniFrac plot highlighted compositional differences. S1 and S2 consistently stood out in both metrics, reflecting their uniqueness in presence/absence and relative abundance of features (Figure 6 and Figure 7). They showed strong separation, indicating unique ecological niches or significant variation in community structure compared to the others. The soil clustering of intercropping and green manure bacteriomes suggested shared ecological or environmental factors influencing their microbial compositions.

## 5. Conclusions

The presence of crop and root exudates caused a decrease in soil pH compared to the fallow soil. Intercropping boosts soil solute concentration and increases available N and K with respect to the cultivation of oats alone. The structure of bacteriome in soil rhizosphere and green manure highly differs from that of bulk soil, highlighting the importance of crops as a whole and mixed cultivation in particular for soil biodiversity, structure and nutrition. Although Proteobacteria were the most represented phylum in the present study, their importance decreased in the cultured variants, and the proportion of Actinobacteria increased. The presence of plants was crucial for the bacterial community structure, showing similarities among these bacteriomes and significant dissimilarities with control variants, without plants. Our understanding is that these differences resulted from the changes in physicochemical properties, thus contributing to the improved bacteriome biodiversity. Moreover, the intercropping of oats with vetch influenced the diversity of soil bacteriome and contributed significantly to the species number and richness. The observed trends suggest that oats–vetch intercropping boosts microbial diversity and enhances the functional capacity of soil agroecosystems. Future research should focus on identifying specific microbial taxa driving important benefits and exploring their roles in soil nutrient dynamics and plant health.

## Figures and Tables

**Figure 1 microorganisms-13-00977-f001:**
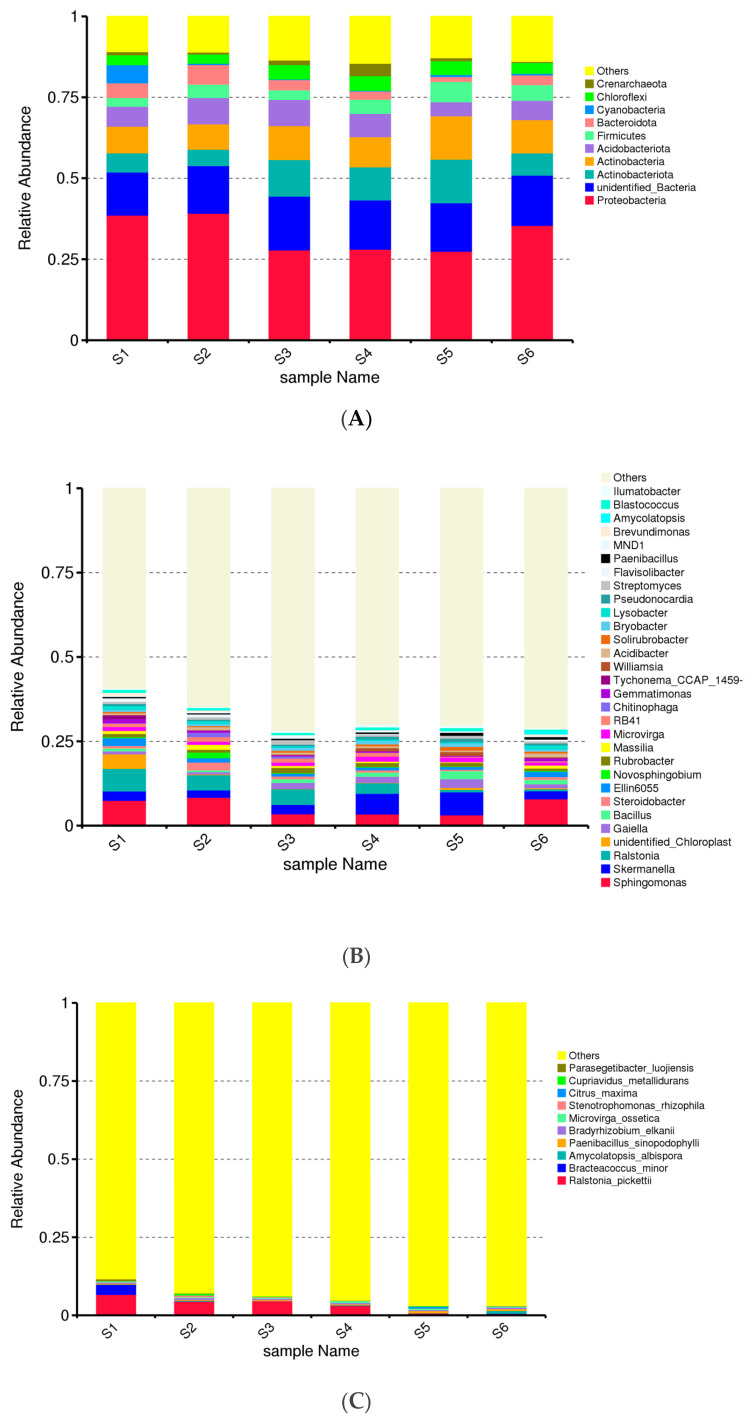
Relative abundance of prokaryotic taxa at the phylum (**A**), genus (**B**) and species (**C**) levels. The graphs show the distribution of microbial taxa, providing a broad overview of the dominant prokaryotic groups present across different samples. Each graph represents a major category of prokaryotes and its relative abundance within the soil bacteriome for each sample (before sowing: S1—bulk soil; ripening phase: S2—bulk soil, S3—oats rhizosphere, S4—intercropping rhizosphere; 3 months after green manure: S5—intercropping field, S6—bulk soil).

**Figure 2 microorganisms-13-00977-f002:**
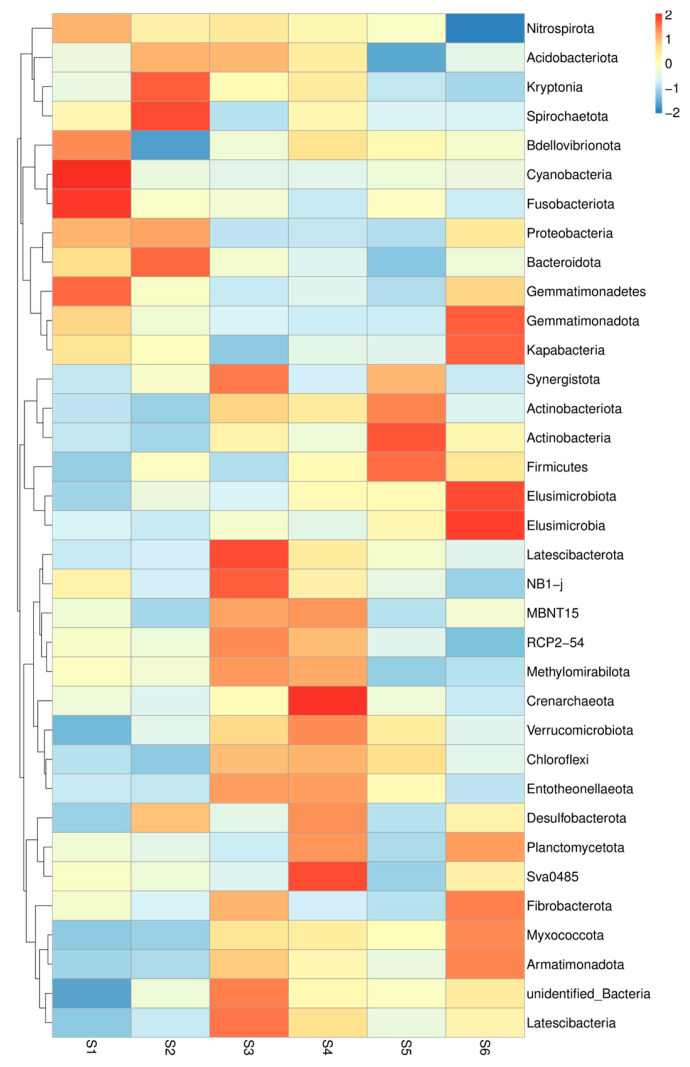
Heatmap of the relationship between treatments and rhizosphere bacteria. The heatmap visualizes the relationships between different rhizosphere bacterial communities across the soil samples. The colors represent the degree of similarity or dissimilarity in bacterial composition between samples, with warmer colors indicating higher similarity and cooler colors indicating greater differences (before sowing: S1—bulk soil; ripening phase: S2—bulk soil, S3—oats rhizosphere, S4—intercropping rhizosphere; 3 months after green manure: S5—intercropping field, S6—bulk soil).

**Figure 3 microorganisms-13-00977-f003:**
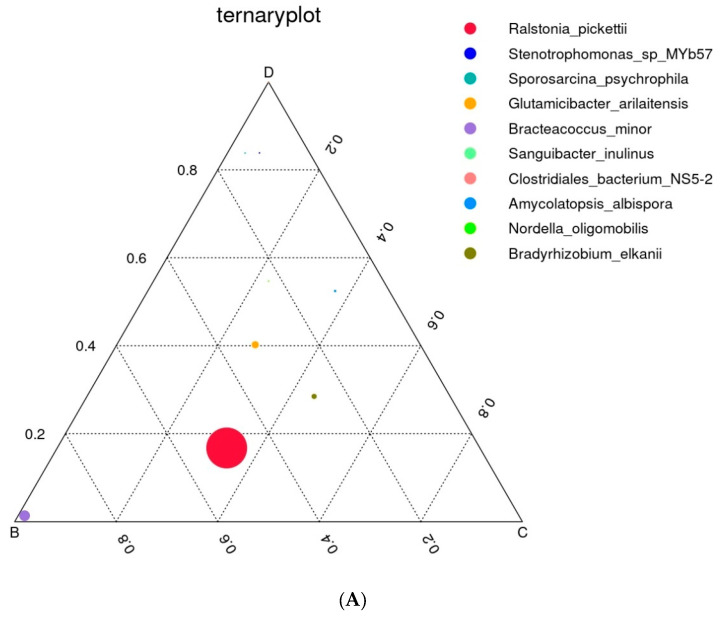
Triple distribution diagram of dominant prokaryotic types. (**A**) The graph shows a triple distribution diagram illustrating the dominant prokaryotic types across three phases: B—S2, C—S3, and D—S4. This diagram highlights the variation and dominance of specific prokaryotic taxa at different cultivation management systems and the control (bulk soil). (**B**) Flower (Venn) diagram of feature overlap with the overlap and unique features of prokaryotic communities across six sample groups (before sowing: S1—bulk soil; ripening phase: S2—bulk soil, S3—oats rhizosphere, S4—intercropping rhizosphere; 3 months after green manure: S5—intercropping field, S6—bulk soil).

**Figure 4 microorganisms-13-00977-f004:**
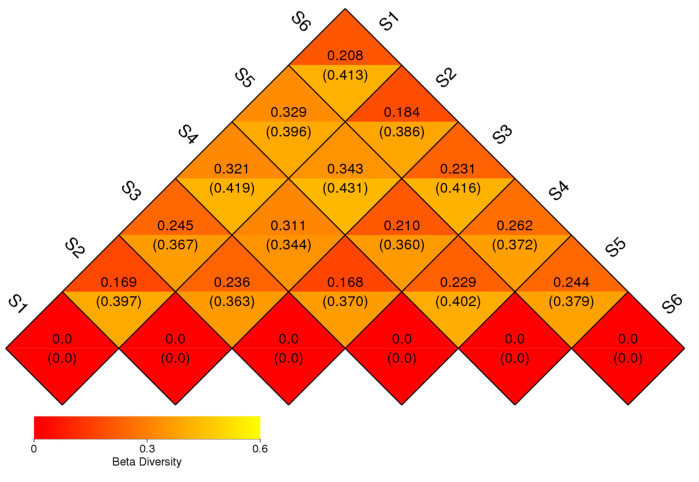
Beta diversity of prokaryotic communities’ samples. It shows how the communities differ from one to another, with samples that are more similar grouped together and those with greater differences placed farther apart. The heatmap highlights the variation in microbial composition between the samples. The numbers inside the parentheses refer to the UniFrac-unweighted analysis, while those shown outside refer to the UniFrac-weighted analysis (before sowing: S1—bulk soil; ripening phase: S2—bulk soil, S3—oats rhizosphere, S4—intercropping rhizosphere; 3 months after green manure: S5—intercropping field, S6—bulk soil).

**Figure 5 microorganisms-13-00977-f005:**
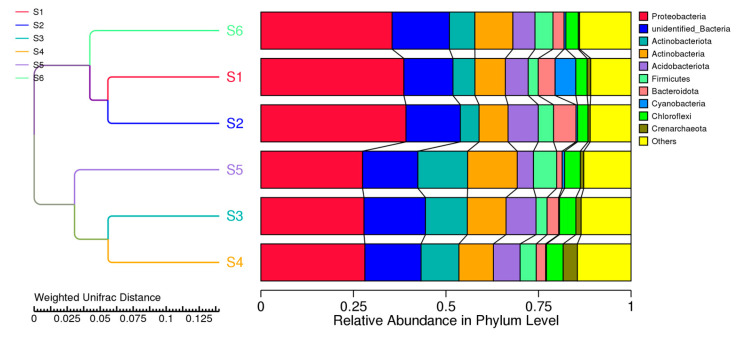
UPGMA cluster tree based on weighted UniFrac distances, which measure the phylogenetic similarity between prokaryotic communities. The tree groups samples are based on their microbial composition, with closer branches indicating more similar communities (before sowing: S1—bulk soil; ripening phase: S2—bulk soil, S3—oats rhizosphere, S4—intercropping rhizosphere; 3 months after green manure: S5—intercropping field, S6—bulk soil).

**Figure 6 microorganisms-13-00977-f006:**
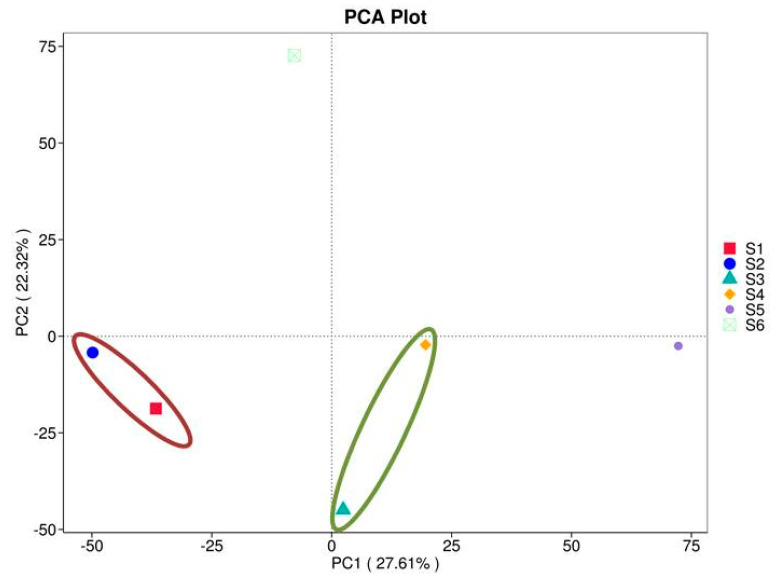
PCA of prokaryotes from the studied soils (before sowing: S1—bulk soil; ripening phase: S2—bulk soil, S3—oats rhizosphere, S4—intercropping rhizosphere; 3 months after green manure: S5—intercropping field, S6—bulk soil). The axes represent the principal components that capture the most variation in the bacteriome composition data. Samples that are closer together in the plot have similar microbial compositions, while those farther apart show greater differences. The ellipse showed clustering of samples grouping with respect to PC1 and PC2.

**Figure 7 microorganisms-13-00977-f007:**
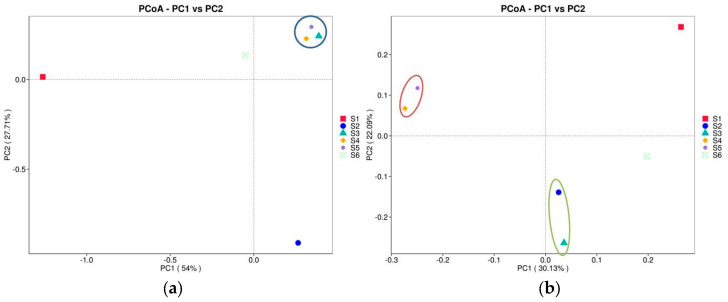
The PCoA plots show the relationships among the studied soils based on two distance metrics: weighted UniFrac (**a**) and unweighted UniFrac (**b**). These plots visualize how microbial communities differ between samples (before sowing: S1—bulk soil; ripening phase: S2—bulk soil, S3—oats rhizosphere, S4—intercropping rhizosphere; 3 months after green manure: S5—intercropping field, S6—bulk soil). The weighted UniFrac metric accounts for both the presence and abundance of microbial taxa, while the unweighted UniFrac metric focuses on the presence or absence of taxa, ignoring their abundance. The ellipse showed clustering of samples grouping with respect to PC1 and PC2.

**Table 1 microorganisms-13-00977-t001:** Origin of soil samples by presence or absence of plant and time point.

	March (Before Sowing)	June (Ripening)	October (3 Months After Green Manure)
Bulk soil	S1	S2	S6
Rhizosphere soil		S3 (oats) S4 (intercropping)	S5 (green manure)

**Table 2 microorganisms-13-00977-t002:** Physicochemical characteristics of the soil samples on a dry weight basis. Data show mean and standard error (n = 3) (before sowing: S1—bulk soil; ripening phase: S2—bulk soil, S3—oats rhizosphere, S4—intercropping rhizosphere; 3 months after green manure: S5—intercropping field, S6—bulk soil). Different letters show significant difference among the studied soil samples (*p* < 0.05).

	Clay (%)	pH (H_2_O)	EC (μS/cm^−1^)	Total N (mg/kg)	N-NH_4_ (mg/kg)	N-NO_3_ (mg/kg)	Total P (P_2_O_5_, mg/kg)	Total K (K_2_O, mg/kg)	Organic C (g/kg)
S1	31.25 ± 1.12	7.78 ab	189.38 a	20.51 a	12.20 a	8.30 a	2.53 b	150.20 a	9.3 b
S2	-	7.81 a	143.95 c	11.20 d	6.40 e	4.80 d	3.56 a	97.50 c	7.55 b
S3	-	7.67 c	126.15 d	16.20 c	9.29 cd	6.91 bc	3.77 a	105.25 c	11.28 a
S4	-	7.72 bc	181.75 b	19.47 ab	11.33 ab	8.15 ab	2.34 b	131.68 b	8.80 b
S5	-	7.66 d	114.88 e	17.05 bc	10.48 bc	8.55 a	3.93 a	128.28 b	12.45 a
S6	-	7.79 ab	191.00 a	15.83 c	8.98 d	6.85 c	2.14 b	127.08 b	8.55 b

**Table 3 microorganisms-13-00977-t003:** A comprehensive summary of the sequencing results from the soil samples collected depending on time point and crop cultivation system (before sowing: S1—bulk soil; ripening phase: S2—bulk soil, S3—oats rhizosphere, S4—intercropping rhizosphere; 3 months after green manure: S5—intercropping field, S6—bulk soil).

Prokaryotes	RawPE	Combined	Qualified	Nochime	GC, %	Q20, %	Q30, %	Efficiency, %
S1	130,849	129,701	127,816	104,967	55.98	98.27	94.31	80.22
S2	139,942	138,856	136,603	116,898	55.67	98.27	94.31	83.53
S3	143,740	142,454	139,950	119,471	56.82	98.15	94.09	83.12
S4	128,339	127,264	125,098	104,163	56.64	98.22	94.24	81.16
S5	136,358	135,040	132,479	113,834	57.08	98.05	93.86	83.48
S6	135,771	134,592	132,188	111,853	56.35	98.09	93.89	82.38

Nochime: removed chimera sequence represents the clean dataset after chimera filtering.

**Table 4 microorganisms-13-00977-t004:** Table summarizes the alpha diversity indices, which measure the richness and evenness of microbial communities in the soil samples. These indices provide insights into the overall diversity within each sample, helping to assess the complexity and health of the soil bacteriome under different agricultural practices (before sowing: S1—bulk soil; ripening phase: S2—bulk soil, S3—oats rhizosphere, S4—intercropping rhizosphere; 3 months after green manure: S5—intercropping field, S6—bulk soil). Different letters show significant difference among studied soil samples (*p* < 0.05).

	Observed_Species	Shannon	Simpson	Chao1	ACE	Goods_Coverage	PD_Whole_Tree
S1	3307 d	9.109 e	0.990 b	3550.099 f	3553.298 e	0.995 a	252.180 bc
S2	3483 c	9.351 d	0.994 ab	3676.867 e	3712.133 d	0.995 a	253.491 bc
S3	3513 c	9.591 b	0.995 a	3802.227 c	3793.272 c	0.995 a	246.634 c
S4	4003 a	9.590 b	0.994 ab	4286.903 a	4424.361 a	0.993 a	285.315 a
S5	3486 c	9.438 c	0.994 ab	3706.513 d	3719.565 d	0.995 a	257.743 b
S6	3663 b	9.758 a	0.996 a	3867.099 b	3910.527 b	0.995 a	279.505 a

## Data Availability

The data records will be accessible using the following link: https://www.ncbi.nlm.nih.gov/sra/PRJNA1190667, accessed on 21 April 2025.

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
