# Peer review of "Intercropping of Oats with Vetch Conducts to Improve Soil Bacteriome Diversity and Structure"

_microorganisms, 2025, doi:10.3390/microorganisms13050977_

Round 1
Reviewer 1 Report (Previous Reviewer 3)
Comments and Suggestions for Authors
I have carefully reviewed this re-submitted manuscript. All questions raised previously have been well addressed.
In this re-submitted manuscript, there is only one new question that should be solved by authors, namely, which soil physicochemical properties drive the changes in soil bacterial community diversity and structure.
Author Response
Dear Reviewer,
Thank you for all your efforts in revising the manuscript, thus improving its quality. The peer-review process is essential for that reason. In that sense, your question is very well-targeted. We have commented on this subject in the discussion and included it in the conclusion. The changes in pH among bulk and rhizosphere soil and the increased potassium and nitrogen concentrations in variants with intercropping were highlighted, where the diversity was increased compared to the variant with growing oats alone. However, your question is pertinent, as the topic could be explained better. That is why I included some sentences for reinforcement in the discussion and in the conclusions (lines 624 and 744).
Reviewer 2 Report (Previous Reviewer 2)
Comments and Suggestions for Authors
The manuscript entitled “Intercropping of oats with vetch conducts to improved soil bacteriome diversity and structure” is devoted to soil depletion during intensive crop cultivation. Authors investigated the effect of growing oats alone or together with vetch and green manure on the structure of the rhizosphere bacteriome. To my mind this manuscript is corresponding to the aims and scopes of the “Microbiomes” journal. I am ready to recommend it for publication because the authors made significant changes to the text and took into account all my comments. In this form, the manuscript became more logically structured and the conclusion looks worthy and shows the prospects of the important data obtained by the authors for further application in practice.
Author Response
Dear Reviewer,
On behalf of the authors, I would like to express our gratitude for all your efforts to improve the quality of the research presentation and the manuscript as a whole.
Kind regards,
Stefan Shilev
Reviewer 3 Report (New Reviewer)
Comments and Suggestions for Authors
The reviewed article presents results of research on impact of the field cultivation methods and the type of cultivated plants on the structure of the rhizosphere bacteriome in soil. The presented data are novel and quite interesting, and are relevant to the field of the journal. They are presented in a well-structured manner. The references are relevant and quite new (32% of cited publications are from the last 5 years, most from the last 25 years).
The goal of research was to determine the effect of growing oats alone vs. together with vetch and green manure on the structure of the bacterial composition of rhizosphere in soil.
The main concern is that the experimental design is not described well enough to determine if it is appropriate for testing the hypothesis about the impact of the field cultivation methods and the type of cultivated plants on bacterial community composition. According to the Materials and Methods section (lines 197-218) the experiment was carried out at the Training and Experimental Field of the Agricultural University of Plovdiv on alluvial-meadow soil. The soil in the area is characterized by huge diversity in terms of its type. It is not mentioned how big the field is. Moreover, it was then divided into three parts with three different treatments. Rhizosphere soil samples were taken from each of the three replicates from 0-20 cm depth on six different occasions. That would mean that only one soil sample (in 3 replicates) was taken for every treatment and for every occasion from the whole field (18 in total, please see lines 216-217). Considering the soil diversity in the area, it is uncertain that it can be used to confirm the hypothesis of the impact of the treatment methods and plants that were grown on the field on the rhizosphere bacteriome. There is no more information about the selected sampling points or justification, why those locations were chosen for analysis.
The figures and tables are very nicely prepared and easy to understand. Unfortunately, there are no explanations for abbreviations of letters a, b, c in Table 2. There is also no explanation for word ‘Nochime’ in Table 3.
Additional small editorial errors:
- Lines 266, 273, 275: references are cited incorrectly (year of publication is not necessary),
- Line 467: it should be Paenibacillus, not Peanibacillus
- Line 501: is should be species, not speciec
Supplementary Material was not available for review.
Author Response
Dear Reviewer,
Thank you very much for the valuable comments. We believe they are very important to improve the paper’s quality. Concerning your comments:
Comment 1: The main concern is that the experimental design is not described well enough to determine if it is appropriate for testing the hypothesis about the impact of the field cultivation methods and the type of cultivated plants on bacterial community composition. According to the Materials and Methods section (lines 197-218) the experiment was carried out at the Training and Experimental Field of the Agricultural University of Plovdiv on alluvial-meadow soil. The soil in the area is characterized by huge diversity in terms of its type. It is not mentioned how big the field is. Moreover, it was then divided into three parts with three different treatments. Rhizosphere soil samples were taken from each of the three replicates from 0-20 cm depth on six different occasions. That would mean that only one soil sample (in 3 replicates) was taken for every treatment and for every occasion from the whole field (18 in total, please see lines 216-217). Considering the soil diversity in the area, it is uncertain that it can be used to confirm the hypothesis of the impact of the treatment methods and plants that were grown on the field on the rhizosphere bacteriome. There is no more information about the selected sampling points or justification, why those locations were chosen for analysis.
Response: Dear Reviewer, thank you for the specific and accurate comment. As this is the second revision, the Materials and Methods section has already been improved. We agree with your comment concerning the lack of detailed description of plots and experimental area, thus it was included: Each replicate was located on a plot of 35 m2, a total of 105 m2 per treatment. This clarification was inserted in line 149 of the Materials and Methods section. In this sense, the area of ​​each plot is large enough to provide conditions close to real production and not too high to raise doubts about the homogeneity of the tested soil (35 m2 x 3 repetitions x 3 variants = 315 m2 + guard zone between repetitions and variants - 400 m2 in total). As we mentioned in the manuscript, there is a diversity of soils in the area, but their location is known and far from the current experimental area. The sampling plants in each replicate were randomly selected at each sampling time (inserted in line 161). Considering that the NGS analyses are very precise and accurate, we believe that using three replicates per variant is not an obstacle to obtaining reliable results. This has also been established in other similar studies.
Comment 2: The figures and tables are very nicely prepared and easy to understand. Unfortunately, there are no explanations for abbreviations of letters a, b, c in Table 2. There is also no explanation for the word ‘Nochime’ in Table 3.
Response: Thank you for drawing our attention to these. We added clarification to the title of Table 2. The term “Nochime” refers to the dataset after removing chimeric sequences during quality control. We have now included a clarification in the table caption for Table 3, explaining that the removed chimaera sequence (Nochime) represents the clean dataset after chimaera filtering.
Comment 3: Lines 266, 273, 275: references are cited incorrectly (year of publication is not necessary)
Response: Dear Reviewer, thank you for this comment. The error was fixed.
Comment 4: Line 467: it should be Paenibacillus, not Peanibacillus
Response: Thank you for the comment. The error was fixed.
Comment 5: Line 501: is should be species, not speciec
Response: Thank you for the comment. The error was fixed.
Comment 6: Supplementary Material was not available for review.
Response: Dear Reviewer, we added the material to the system. Now I am enclosing it again. Thank you for the clarification.

Round 2
Reviewer 3 Report (New Reviewer)
Comments and Suggestions for Authors
In the reviewed article "Intercropping of oats with vetch conducts to improved soil bacteriome diversity and structure" Authors presented research on the effect of growing oats alone or together with vetch and green manure on the structure of the rhizosphere bacteriome. The authors have responded satisfactorily and substantively to all the points and comments made in the previous review. The article is written well, the presented data are interesting and novel.
This manuscript is a resubmission of an earlier submission. The following is a list of the peer review reports and author responses from that submission.
Round 1
Reviewer 1 Report
Comments and Suggestions for Authors
The manuscript addresses a significant topic by exploring the effects of intercropping oats with vetch on soil microbial diversity and structure. It presents well-structured experiments utilizing 16S rRNA metagenomics to analyze soil microbiomes. The strengths of the manuscript include its clear objectives, appropriate use of metagenomics, and comprehensive analysis of microbial diversity. However, the manuscript has several areas that need improvement, including language, methodological details, data presentation, and reference alignment.
1. While the introduction explains the individual roles of oats and vetch (page 2), it could emphasize their combined benefits in intercropping and include references demonstrating the synergistic effects of cereal-legume intercropping on forage yield and soil quality.
2. Recent literature supports the specification of how legumes like vetch contribute to microbial diversity through nitrogen fixation and root exudates.
3. Ensure all statements about ecological impacts or microbiomes are supported by appropriate citations.
4. Focus on highlighting the study's novelty and directly connect the introduction's conclusion to the research objectives.
5. The description of the experimental design is adequate, but further details on soil type classification (FAO classification) should be justified with regional relevance.
6. The methods for analyzing pH, EC, and nutrient content are adequately described, but references for the specific methods are not included.
7. The DNA extraction method is not clearly described as following "the manufacturer’s instructions." The specific kit or manufacturer name should be included for reproducibility. Provide detailed information about the DNA extraction kit, the amount of soil used per sample, and any modifications made to the protocol.
8. To strengthen the analysis, include statistical comparisons of soil parameters across treatments. For example, highlight significant differences between intercropped soils and controls.
9. Beta diversity results are briefly mentioned but do not explain the ecological implications of observed differences between sample groups.
10. The description of dominant phyla (e.g., Proteobacteria, Actinobacteria) is informative but general. The functional roles of these groups in nutrient cycling or plant health are not sufficiently discussed (page 6). Include hypotheses or references to similar studies to explain the suppression of pathogens.
11. Figures 1, 2, and 6 (pages 6–12) are visually clear but lack detailed captions explaining what the data represent. For instance, Figure 1B (relative abundance at the genus level) could benefit from highlighting key genera affected by intercropping. Enhance figure captions with brief descriptions of the biological relevance of the data.
12. While the manuscript mentions statistical tests, the results are not explicitly stated for many comparisons. For instance, are differences in alpha or beta diversity indices statistically significant? To provide statistical support for claims, include p-values or confidence intervals for key results.
13. The manuscript cites some studies to support its findings but lacks critical comparisons to previous research. For example, the findings on microbial diversity (page 14, lines 363–369) are not well-aligned with the broader body of literature on cereal-legume intercropping.
14. Make the conclusion more impactful by emphasising how these findings contribute to sustainable agriculture and microbial ecology. Conclude with a clear and specific statement about the significance of the findings for sustainable agriculture and microbial ecology.
15. Provide deeper mechanistic explanations for key findings, such as pathogen suppression and enhanced microbial diversity.
Reviewer 2 Report
Comments and Suggestions for Authors
The manuscript entitled « Intercropping of oats with vetch conducts to improved soil bacteriome diversity and structure» is devoted to the effects of oat-vetch intercropping on the soil bacterial microbiome and enzyme activities. This study provides a theoretical basis for understanding the microbial communities associated with oat–legume intercropping. It offers valuable guidance for managing soil and plant health by reducing pathogens and enhancing plant productivity. To my mind this manuscript is corresponding to the aims and scopes of the “Microorganisms” journal. I am ready to recommend it for publication after some corrections.
1. The methodological features of the work should be removed from the abstract
2. L90 change Obrazets to sample
3. The process of sample selection should be described in detail
4. It is necessary to write the sample weight from which DNA was isolated
5. The font size of the legend in Figure 1 should be increased
6. Figure 2 should be moved to the supplementary
7. Phrases such as Principal component analysis (PCA), and component of beta-diversity analysis should be removed from the entire text. They can only be used in materials and methods.
8. The conclusion should be significantly expanded. The authors obtained interesting results, which, unfortunately, were lost in general phrases. Try to make a conclusion based on specific results.
Reviewer 3 Report
Comments and Suggestions for Authors
I have carefully reviewed the manuscript entitled “Intercropping of oats with vetch conducts to improved soil bacteriome diversity and structure” submitted by Petkova et al. to a MDPI journal Microorganisms. The objective of this study was to assess the effects of oat-vetch intercropping on soil bacterial microbiome and enzyme activities. The field experiment design was correct, but the testing method was incorrect, especially that the soil samples used for sequencing did not have biological replicates, resulting in unreliable sequencing results. Therefore, I suggest rejecting this manuscript. Some comments are as the following:
1. Line 96. Changing “Soil rhizosphere samples” with “Rhizosphere soil samples”.
2. Line 96-100. Adding a flow chart/image that describes sampling time and location. This will be more intuitive.
3. Only one season of field experiment was conducted, and the stability of the test results needs to be considered seriously.
4. Line 113. Changing “Metagenomic analysis” with “High-throughput amplicon sequencing”.
5. Line 100-101 and Line 175-177. Only six composite soil samples were sequenced (S1 to S6). This means that there is no biological duplication of soil samples, and therefore the analysis involved in sequencing are unreliable (lack of statistical testing). This is a fatal problem.
Comments on the Quality of English Languagecan be improved